# Immunomodulatory Activity of Punicalagin, Punicalin, and Ellagic Acid Differs from the Effect of Pomegranate Peel Extract

**DOI:** 10.3390/molecules27227871

**Published:** 2022-11-15

**Authors:** Miodrag Čolić, Dušan Mihajlović, Marina Bekić, Milan Marković, Branka Dragišić, Sergej Tomić, Nataša Miljuš, Katarina Šavikin, Ranko Škrbić

**Affiliations:** 1Medical Faculty Foča, University of East Sarajevo, 73300 Foča, Bosnia and Herzegovina; 2Serbian Academy of Sciences and Arts, 11000 Belgrade, Serbia; 3Institute for the Application of Nuclear Energy, University of Belgrade, 11000 Belgrade, Serbia; 4Faculty of Medicine, University of Banja Luka, 78000 Banja Luka, Bosnia and Herzegovina; 5Institute for Medicinal Plant Research “Dr. Josif Pančić”, 11000 Belgrade, Serbia

**Keywords:** pomegranate, ellagitannins, lymphocytes, culture, cytokines

## Abstract

Background: Our recent study has shown that pomegranate peel extract (PEx) showed significant immunomodulatory activity, which might be caused by ellagitannins. The aim of this work was to test the hypothesis that ellagitannin components act synergistically in the modulation of cytokine production. Methods: Human peripheral blood mononuclear cells (PBMCs) from healthy donors were stimulated with phytohemagglutinin and treated with different concentrations of PEx or punicalagin (PG), punicalin (PN), and ellagic acid (EA), alone or with their combinations. Cytotoxicity, cell proliferation, and cytokine production were determined. Results: Non-cytotoxic concentrations of all compounds significantly inhibited cell proliferation. IC50 values (μg/mL) were: EA (7.56), PG (38.52), PEx (49.05), and PN (69.95). PEx and all ellagitannins inhibited the levels of TNF-α, IL-6, and IL-8, dose-dependently, and their combinations acted synergistically. PEx and all ellagitannins inhibited Th1 and Th17 responses, whereas the lower concentrations of PEx stimulated the production of IL-10, a Treg cytokine, as did lower concentrations of EA. However, neither component of ellagitannins increased Th2 response, as was observed with PEx. Conclusions: The combination of PG, PN, and EA potentiated the anti-inflammatory response without any significant synergistic down-modulatory effect on T-cell cytokines. The increased production of IL-10 observed with PEx could be attributable to EA, but the examined ellagitannins are not associated with the stimulatory effect of PEx on Th2 response.

## 1. Introduction

Pomegranate (Punica granatum) is very rich in various phytochemicals with significant health-promoting properties and beneficial effects in different pathologies, such as diabetes, cardiovascular diseases, and chronic inflammatory and degenerative conditions, including those of autoimmune nature, cancer, and many others [1,2,3]. Most of them are good anti-microbials [4]. Almost all components of pomegranate (seed, juice, peel, and leaf) contain bioactive compounds. Among them, the peel is of special importance due to main two reasons. It is a non-edible part of the fruit and it is usually treated as waste. On the other side, the peel is richest in compounds with anti-oxidant activities in comparison to other parts of the fruit [5,6]. The most important phytochemicals in the peel of pomegranate are different polyphenols. Among them, tannins are specific and contribute to more than 90% of the total anti-oxidant components of pomegranate [7,8]. Tannins (dominantly ellagitannins and gallotannins) are very hydrolyzable and represent the specific polyphenolic signature of pomegranate [9]. Other polyphenols present in significantly lower amounts include hydroxybenzoic derivatives (gallic acid), hydroxycinnamic acid derivatives (different cinnamic acids), flavonols (quercetin and its glycosides, kaempferol, and rutin), flavones (apigenin and luteolin), flavanones (naringenin and hesperidin), isoflavones (phytoestrogens), anthocyanins and proanthocyanidins (catechin and epicatechin) [10]. Other components of the peel are minerals, sugars, fatty acids, organic acids, crude fibers, and alkaloids (mostly pelletierines) [7,8].

Punicalagin (PG), punicalin (PN), ellagic acids (EA), and their metabolites urolithins exert anti-oxidative and anti-inflammatory properties, as shown in numerous in vitro and in vivo experiments [1,2,11]. The suppression of inflammation is characterized by the down-regulation of different pro-inflammatory cytokines, chemokines, derivatives of arachnoid acids, and many inflammatory mediators, and most of these effects can be associated with the reduction of oxidative stress and modulation of autophagy and apoptosis. The anti-inflammatory effect of pomegranate extract (PEx) was confirmed in different autoimmune experimental models in vivo [11,12,13,14,15] and in humans [16,17,18,19,20]. The investigated parameters, such as the production of pro-inflammatory cytokines, NO, prostaglandin E2 (PGE2), or M1/M2 macrophage polarization, depended on the goal of the studies [1,2,21,22]. The immunomodulatory activity of ellagitannins is the least-studied of these parameters, and both immunosuppression and immunostimulation were described [4]. In this context, polyphenols have been shown to regulate intestinal mucosal immunity, allergic diseases, and antitumor immune responses [23]. Except for ellagitannins and gallagic acid from PEx, which show anti-oxidative and immunomodulatory properties [4,7,8], novel research has shown that anthocyanins, alkaloids, and some fatty acids also contribute to the biological effects. Other polyphenols from pomegranate may be involved in the modulation of immune responses through their binding to cellular receptors. Different immune cells express multiple types of polyphenol receptors, which are responsible for the recognition and cellular uptake of polyphenols, which subsequently activate signaling pathways to different biologically active processes including the triggering of immune responses [23]. The combination of these compounds may exert synergistic effects that are significantly higher than the effect of single compounds. However, such a presumption was not confirmed.

In our recent paper [24], we have shown that pomegranate peel extract (PEx) suppressed the production of Th1 (IFN-γ), Th17 (IL-17A, IL-17F, and IL-22), Th9 (IL-9), and proinflammatory cytokines (TNF-α and IL-6) in culture supernatants of human peripheral blood mononuclear cells (PBMC) stimulated with phytohemagglutinin (PHA), a potent T-cell mitogen. Lower concentrations upregulated Th2 (IL-5 and IL-13) and Treg (IL-10) responses, as well as CD4+CD25hiFoxp3+ cell frequency. Higher concentrations of PEx increased the frequency of IL-10- and TGF-β-producing T-cells. All these data suggest the complex immunoregulatory effects of PEx on T cells, which could be beneficial in the suppression of chronic inflammatory and autoimmune diseases. Since the composition of PEx is very complex and at least three ellagitannin components (PG, PN, and EA) dominated over other polyphenols, in this work, we tested the hypothesis that these components acted synergistically in modulating cytokines involved in inflammatory and immune responses. Therefore, this was the main aim of this study performed in the same vitro culture model on human PBMC stimulated with PHA.

## 2. Results

### 2.1. Cytotoxicity of Ellagitannin Components from PEx

In order to investigate the immunomodulatory role of ellagitannin components from PEx, it is important to determine non-cytotoxic concentrations. Therefore, the cytotoxicity experiment of PEx and their main constituents on PBMC was the first part of this study. Different concentrations of PEx, PG, PN, and EA, ranging from 1–320 μg/mL were applied in PBMC cultures and cytotoxicity was determined by the MTT test. The results presented in Figure 1 show that PEx, PG, and PN are cytotoxic at concentrations of 80 μg/mL and higher, whereas the cytotoxicity of EA started from the concentration of 20 μg/mL. The cytotoxic IC_50_ values (mean values of 5 donors) (μg/mL) were as follows: PEx: 109.5; PG: 91.07; PN: 113.4; EA: 43.43. 

### 2.2. Proliferative Activity of PBMC by the Influence of Ellagitannin Components from PEx 

The production of cytokines by activated PBMC depends on the number of cells in the culture. Therefore, the next experiment was set up in order to study the effect of different concentrations of PEx, PG, PN, and EA on the proliferative activity of PBMC, stimulated with PHA. The results are expressed as relative proliferation in comparison with control (only PHA-stimulated PBMC cultures). PEx and PN were inhibitory at concentrations of 40 μg/mL and higher, PG was inhibitory at 20 μg/mL and higher, whereas inhibition of PBMC in the presence of EA started from 2.5 μg/mL (Figure 2). The IC_50_, determined on the basis of relative proliferation (%) relative to control was as follows: PEx: 49.05; PG: 38.52; PN: 69.95; EA: 7.56 (All results are expressed as μg/mL). In Figure 3 representative histograms with proliferation indices (PI) and division indices (DI) are shown.

### 2.3. Effect of Ellagitannin Components and Their Combinations on the Production of Proinflammatory Cytokines in PBMC Cultures

Three proinflammatory cytokines (TNF-α, IL-6, and IL-8) were analyzed in culture supernatants (Figure 4). Based on the cytotoxic potential of the components PEx, PG, and PN were investigated at double increasing concentrations ranging from 10–80 μg/mL, whereas EA was used at concentrations (1, 2.5, 5, and 10 μg/mL). Based on preliminary results two combinations of PG, PN, and EA were used. Combination 1 with lower inhibitory concentrations of ellagitannins: (PG 10 μg/mL, PN 20 μg/mL, and EA 1 μg/mL); Combination 2 with higher inhibitory concentrations of ellagitannins: (PG 20 μg/mL, PN 40 μg/mL, and EA 2.5 μg/mL). The effect of ellagitannin combinations was compared with the corresponding single components in order to see if there are any synergistic or additive effects.

Although the cytokines exerted similar activity, the influence of ellagitannin components was different. The values of TNF-α showed that PEx and all ellagitannin components exerted a dose-dependent inhibitory effect. The inhibition of TNF-α level in the presence of PG was seen at the concentration of 20 μg/mL and higher and showed a very similar inhibitory pattern as did PEx. PN showed similar inhibition at double higher concentrations compared to PG and PEx. The inhibitory effect of EA was strongest and started from 2.5 μg/mL.

When non-inhibitory concentrations of PG, PN, and EA were combined (combination 1), a statistically significant inhibition was observed, indicating that these components acted synergistically in decreasing TNF-α concentrations. When combination 2 was applied the difference was seen in the comparison with the same concentration of PG (20 μg/mL) and PN (40 μg/mL) but not in comparison with EA (2.5 μg/mL).

The inhibitory effect of PEx (40 and 80 μg/mL), PG (40 and 80 μg/mL), and PN (80 μg/mL) on IL-6 production was lower, compared to TNF-α. The inhibitory dose-dependent effect of EA (ranging from 2.5–10 μg/mL) was similar to that seen with TNF-α. When combinations of ellagitannins were applied, a synergistic inhibitory effect on IL-6 production was seen only with combination 2. The inhibitory activity of PEx and all ellagitannin components on IL-8 production was similar to that seen with TNF-α. However, synergistic inhibition was seen only with combination 2, as was observed with IL-6.

### 2.4. Effect of Ellagitannin Components and Their Combinations on the Production of T-Cell Subset Cytokines in PBMC Cultures

Four T-cell subset cytokines were analyzed in culture supernatants of PHA-activated PBMC: IFN-γ (Th1), IL-5 (Th2), IL-17A (Th17), and IL-10 (Treg). The results are presented in Figure 5. The production of IFN-γ was significantly down-regulated, dose-dependently, in the presence of all concentrations of ellagitannin components. The down-modulatory effect of PEx (20–80 μg/mL) was stronger in comparison with the same concentrations of PG. When combinations of ellagitannin were used, no synergistic effect was observed.

The level of IL-17A was significantly lower and dose-dependent in the presence of PG and PEx (20–80 μg/mL). Similarly, as observed with IFN-γ, the effect of PEx was stronger. PN was inhibitory at the highest concentration (80 μg/mL), whereas all concentrations of EA decreased the level of IL-17A, dose-dependently. Both combinations of ellagitannins potentiated the effect of single components but the effect was not recognized as synergistic.

The lowest concentration of PEx (10 μg/mL) increased the production of IL-5, whereas the concentrations of 40 and 80 μg/mL were inhibitory. Similarly, PG (40 and 80 μg/mL), PN (80 μg/mL) and EA (2.5–10 μg/mL) were inhibitory but neither concentration of all these components increased IL-5 production. The concentrations of IL-5 in cultures with combination 2 were higher compared to the corresponding concentration of EA. No other differences were observed.

Lower concentrations of PEx (10 and 20 μg/mL) stimulated IL-10 production, whereas higher concentrations (40 and 80 μg/mL) were inhibitory. Similar effects exerted lower (1 and 2.5 μg/mL) and higher (5 and 10 μg/mL) concentrations of EA. PG at concentrations of 20–80 μg/mL and PN at the highest concentration (80 μg/mL) decreased IL-10 production. Although the effect of both combinations was statistically different in comparison with single components (PN *versus* combination 1; EA *versus* combination 2), no significant additive or synergistic effects were seen.

## 3. Discussion

This study investigated the immunomodulatory properties of different combinations of PG, PN, and EA in order to check whether the effect of PEx on cytokine production by human PBMC culture is a result of synergistic actions of its ellagitannin components. At first, we investigated the cytotoxic potential of these ellagitannins in PBMC culture in order to determine the optimal non-cytotoxic concentrations for the main experiments. We showed that all investigated ellagitannins exerted a cytotoxic effect at higher concentrations as confirmed in our previous study for PEx on PBMC [24] and on different cancer cell lines as demonstrated for PG [25] and EA [26]. PG and PN have been shown to exert a similar dose-dependent cytotoxic effect on preadipocytes and adipocytes as judged by LDH release [27]. In another study, PG was non-cytotoxic on a macrophage line, but this might be due to a much lower concentration (10 μM) applied [28]. According to the IC50, EA exerted the highest cytotoxic potential (IC50 = 43.4 μg/mL), whereas, PN (IC50 = 113.4 μg/mL) showed the lowest effect. It remains to be studied whether the cytotoxicity of ellagitannins depends on used cells and culture conditions because such data are scarce in the literature. For example, the IC50 for EA on mouse macrophages was 23.7 μg/mL [29]. The cytotoxicity of PEx and its constituents could most probably be due to apoptosis induction [24,30].

The level of cytokine production in a culture also depends on cell proliferation activity. We showed that both PEx and the examined ellagitannins showed a strong inhibitory effect on PHA-stimulated PBMC proliferation. The IC50 inhibitory concentrations were much lower than the IC50 cytotoxicity, but the pattern of cell proliferation inhibition was the same (lowest for EA, highest for PN). Although the mechanisms involved in the antiproliferative effect of PEx and their components on lymphocytes have not been studied, an analogy from other cell culture models could be extrapolated. Namely, it has been shown that EA induces a cell cycle arrest in the G0/G1 phase through the activation of the TGF-β/Smad-3 pathway and subsequent down-regulation of cyclins (A2 and E2) due to the up-regulation of cyclin-dependent kinase inhibitor 1A/p21 (CDNK1a/p21) and other kinase inhibitors [31]. PG additionally inhibited cell proliferation of tumor cell lines by blocking the G2/S2 phase of the cell cycle [32]. Cell cycle control by ellagitannins metabolites (EA and urolithins) also involves the modulation of specific micro RNAs through the inhibition of PI3K/Akt–NF-kB and/or MAPK signaling pathways [31]. It can be also hypothesized that in our experiments, the inhibition of the nuclear factor of T-cell activity (NFAT), a key transcription factor for T cell proliferation and T cell differentiation [33], could be involved in the antiproliferative activity of PEx and ellagitannins, based on some similar findings in other cell culture models [34].

Baradaran Rahimi et al., 2020 [1] recently summarized that pomegranate and EA suppress inflammation by down-regulating numerous pro-inflammatory cytokines and chemokines including TNF-α, IL-1β, IL-6, IL-8, and IL-18. In the present study, we showed that all ellagitannins, similarly to PEx, strongly inhibited the production of pro-inflammatory cytokines/chemokines (IL-6, IL-8, and TNF-α), and the individual effect of each component depended on the used concentrations. As could be expected from the cytotoxic and anti-proliferative experiments, EA had the most potent suppressive potential. By using different and well-balanced combinations of these ellagitannins, we showed that they acted synergistically or additively in the suppression of production of all examined pro-inflammatory cytokines, and this is the first report about such a phenomenon. Synergism could be due to the action of individual components on the same target molecules. According to the published data, the target molecules could be numerous, including signaling molecules, receptors, or transcription factors such as cyclooxygenase-2 (COX-2), nuclear factor- kappa-B (NF-κB), Janus kinase (JNK), mitogen-activated protein kinase (MAPK), inducible nitric oxide synthase (iNOS), P38, extracellular signal-regulated kinase (ERK), phosphatidylinositol 3-kinase (PI3K)/AKT/mammalian target of rapamycin (mTOR), PI3k/Akt/mTOR, nuclear factor erythroid 2–related factor 2 (Nrf2), peroxisome proliferator-activated receptor (PPAR)-α and PPAR-γ [31] or protease-activated receptor 2 (PAR2) [35]. These targets were identified by using either PEx extract [36,37,38], EA [39], or PG [40]. It can be supposed that other components from PEx could contribute to the synergism [41]. A number of flavonoids suppress inflammation by down-modulating the expression of proinflammatory cytokines targeting the NF-kB signaling pathway [42].

The most important part of our results was related to the effect of ellagitannins and their combinations on T-cell-producing cytokines and the comparison of their individual and combined effects with the modulating effect of PEx. T helper (Th) cells of the CD4+ phenotype play a central role in the adaptive immune response to pathogens by assisting B cells to produce antibodies, recruiting polymorphonuclear leukocytes to infected sites, and producing cytokines and chemokine to initiate and regulate the immune response [43]. Th cells differentiate into different effectors such as Th1, Th2, Th9, Th17, Th22, Treg (regulatory T cells), and Tfh (follicular helper T cells), upon encountering an antigen presented within the MHC molecules on antigen-presenting cells, dominantly dendritic cells (DCs). Each subset is characterized by different cytokine profiles. Th1 cells produce IFN-γ and TNF-α, which are important for the defense against intracellular bacteria, viruses, and cancer cells. Th2 cells release IL-4, IL-5, and IL-13, which stimulate humoral immunity and anti-helminthic response. Th17 cells produce a number of cytokines from the IL-17 family (cytokines playing an important role in host defense against bacteria, and fungi) and IL-22 (a cytokine important for stabilization of epithelial barriers). Treg secretes IL-10, IL-35 and TGF-β (key immunosuppressive cytokines). [43,44]. This was the reason why we investigated the main representative cytokines of Th1 (IFN-γ), Th2 (IL-5), Th17 (IL-17A), and Treg (IL-10).

There are many examples in the literature that dietary polyphenols down-modulate different T-cell cytokines [22], but the effect of PEx on the complex immunoregulatory cytokine network was not published for pomegranate or individual ellagitannins until our recent study [24]. Namely, we showed that non-cytotoxic concentrations of PEx suppressed the production of Th1 and Th17 cytokines by PHA-activated PBMC, dose-dependently. Lower concentrations of PEx augmented Th2 and Treg responses together with the increased frequency of Treg (CD4+CD25hiFoxp3+ cells), whereas higher concentrations were inhibitory. Such a phenomenon of the PEx effect on the production of IFN-γ, IL-5, IL-17A, and IL-10 by PHA-stimulated PBMC was confirmed in the present study, but differences were observed when the effect of individual ellagitannin components were analyzed. We demonstrated that all investigated individual components inhibited the production of IFN-γ and IL-17A in a similar dose-dependent manner, but no synergism between them was observed. Up to now, a similar cell culture system was not used to study the immunomodulatory activity of pomegranate or ellagitannins, however, different results were published in other cell culture models or by using in vivo experiments. In this context, Stojanović et al., 2017, demonstrated that PEx alleviated symptoms of type 1 diabetes, by inhibiting the infiltration of immune cell into pancreatic islets and decreasing IL-17 and IFN-γ production in gut-associated lymphoid tissue in vivo and in vitro [45]. In another study the treatment with propolis, pomegranate, and grape pomace extracts decreased serum levels of IL-17 in an in vivo rheumatoid arthritis model [46]. Lower concentrations of EA decreased a number of cytokines in tumor cells/PBMC co-cultures but IFN-γ was not affected [47]. Other polyphenols from PEx could be also involved in the down-modulation of IL-17 and IFN-γ production. For example, quercetin inhibited IL-1β, IL-6, IL-8, and IFN-γ in astrocyte cultures [48]. The polyphenols present in blueberry inhibited the production of a number of proinflammatory cytokines, including IL-12 (a key IFN-g-inducing cytokine), in LPS-stimulated RAW264.7 macrophages [49]. Some studies also showed that IL-12 could be a target of polyphenols action and polyphenols-impaired inflammation could be due to the down-regulation of IL-12 production [50]. However, some opposing results have been published. For example, cyclic dimeric ellagitannins have been shown to enhance IFN-γ production by both bovine and human NK cells and T cells [51,52]. In addition, different polyphenols from green tea and red wine have been shown to modulate the immune response by stimulating IL-12 production, and consequently, by promoting the activity of macrophages and cytotoxic cells [53]. Many in vivo experiments with tumor-bearing mice showed that polyphenols increase Th1 response, which is beneficial for anti-tumor immune response [54].

The effect of ellagitannins on Th2 response is less examined. It is generally accepted that polyphenols reduce allergy by downregulating Th2 response [54], a phenomenon that we observed with higher doses of both PEx and individual ellagitannin components. Anderson and Teuber, 2010, showed that walnut kernel polyphenolics and EA inhibited human PBMC proliferation while simultaneously increasing IL-2 production and decreasing IL-13 and TNF-α production. No changes in IL-4 production were observed [55]. In our previous paper [24], we found that the Th2 response (as judged by increased levels of IL-5 and IL-13, but not IL-4) is enhanced in the presence of lower concentrations of PEx, but at higher concentrations, the production of all these cytokines were downregulated. This finding was confirmed in this study by analyzing IL-5 in PEx-treated PBMC cultures. The treatment of mice subjected to contact dermatitis with polyphenols from pomegranate was followed by increasing splenic IL-10-producing and IFN-γ-/IL-4-producing CD4+ T cells [56]. One of the target receptors could be dipeptidyl-peptidase 4 (DPP4), which is inhibited by pomegranate juice, PG, EA and urolithin A [57]. It has been published that DPP4 inhibition leads to up-regulation of the Th2 immune response [58]. However, previous findings contradict our results, as neither of the investigated ellagitannins up-regulated IL-5 production. Some in vivo experiments showed that the treatment of mice with EA significantly reduced the levels of pro-inflammatory cytokines TNF-α, IL-1β, and IL-6 simultaneously by inducing anti-inflammatory cytokines IL-4 and IL-10 [59]. These results imply that maybe other components from PEx could contribute. However, neither of the investigated ellagitannins up-regulated its production. These results imply that other components from PEx could contribute to the Th2-promoting activity. One of the candidates could be alkaloids that show immuno-enhancing activity in different in vitro and in vivo models [60], including up-regulation of Th1 and Th2 cytokines [61].

Another important finding of our results is related to the production of IL-10. As already emphasized, Treg cells produce IL-10, IL-35, and TGF-β, three main immunosuppressive cytokines with pleiotropic functions which play key roles in protecting the host from infection-associated immunopathology, autoimmunity, and allergy [62,63]. We confirmed in this study that PEx, at lower concentrations, augmented the production of IL-10 in PHA-stimulated PBMC cultures. Of the ellagitannins examined, only lower concentrations of EA exerted a similar effect. In this context, our findings are similar to those of Lu et al., 2020 [64], who observed an increased proportion of CD4+Foxp3+ and CD4+ IL-10+ T-cells in the spleen and brain of mice with experimental autoimmune encephalitis treated with PEx. In addition, EA inhibited inflammation by increasing IL-10 production [65]. Other polyphenols from PEx could also contribute to the observed phenomenon. Yahfoufi et al., 2018, showed that lower concentrations of luteolin and quercetin stimulated the expression of IL-10 [66]. In other cell culture models, quercetin and catechins enhanced IL-10 production and inhibited TNFα and IL-1β [67,68]. Quercetin has been shown to reduce the production of IL-17A and IL-21 and to increase IL-10 and TGF-β, which correlate to similar changes in the percentages of Th17+ and Treg cells [69]. This finding is in accordance with the mutual antagonistic effects of these Th subsets [43,44]. Epigallocatechin gallate significantly enhanced the number of Foxp3-positive Treg cells and IL-10 production both in vivo and in vitro by suppressing the NF-kB signaling pathway via inducing epigenetic changes [70].

## 4. Materials and Methods

### 4.1. Study Protocol

This study was performed on human PBMC from healthy volunteers (*n* = 5), both sexes, 28–34 years. The study was approved by the Ethical Committee of the Faculty of Medicine, University of Banja Luka, R. Srpska, Bosnia and Herzegovina, and the participants signed an informed consent. The experiments were performed at the Center for Biomedical Research, Faculty of Medicine, Banja Luka, Institute for the Application of Nuclear Energy, Zemun, Serbia, and the Center for Biomedical Sciences, Medical Faculty Foča, University of East Sarajevo, R. Srpska, Bosnia and Herzegovina.

### 4.2. Pomegranate Peel Extract and Ellagitannin Components

Pomegranate fruits were from a natural locality in the southern region of Bosnia and Herzegovina. The separated peels were dried for 4–6 days at room temperature and grounded in powder. The pomegranate peel extract (PEx) was prepared by the treatment of the powdered peel with 50% ethanol, using 1:10 as a solid to solvent ratio. Extraction was performed in an ultrasonic bath at 600 °C for 40 min. After filtration and evaporation (Büchi R-210 rotary evaporator, Flawil, Switzerland), the total phenolic content in PEx was analyzed spectrophotometrically using the Folin–Ciocalteu method, as previously described [71]. The results were expressed as mg of gallic acid (GA) equivalents per gram of dry weight because GA (0–100 mg/L) was used for the calibration curve. The total content of tannins was expressed as pyrogallol (%, *w*/*w*), according to the European Pharmacopoeia (Ph Eur 7.0 The total flavonoid content was expressed as mg catechin equivalents per gram of sample [72]. All results were presented as the mean of three measurements. The detailed protocol is described in the previous paper [45]. The pomegranate peel was deposited in Botanical Garden “Jevremovac” University of Belgrade (voucher specimen No. BEOU 17742). PG, PN, and EA were obtained from Sigma-Aldrich, St. Louis, MO, USA.

### 4.3. HPLC Analysis

HPLC analyses were performed on the apparatus Agilent 1200 RR HPLC (Agilent, Waldbronn, Germany), equipped with a DAD detector, using reverse-phase analytical column Zorbax SB-C18 (Agilent), as previously described in detail [73]. Orthophosphoric acid in water (1% *v*/*v*) and acetonitrile were mobile phases A and B, respectively. The parameters were as follows. Flow rate: 1 mL/min; gradient elution: 98–90% A (0–5 min); 90% A (5–15 min); 90–85% A (15–20 min); 85–70% A (20–25 min); 70–40% A (25–30 min); 40–0% A (30–34 min); detection—260 and 320 nm. The quantity of investigated compounds in the PEx (PG, PN, and EA) and GA was determined using calibration curves of their standards, and the results were expressed as mg/gram of dry weight. As shown in our previous paper [24] and the Appendix A of this paper, the main compounds of PEx were PG (67.26 ± 0.81 mg/g), PN (31.91 ± 0.22 mg/g), EA (25.11 ± 0.06 mg/g), and GA (9.75 ± 0.05 mg/g).

### 4.4. PBMC Cultures

Blood samples were taken from healthy volunteers. PBMCs were isolated from the blood by Nycoprep (Nycomed, Oslo, Norway) density-gradient centrifugation. Cell viability (higher than 95%) was determined by Trypan blue exclusion. The PBMC samples were adjusted to the concentration of 3 × 10^5^ cells and cultured in 96-well flat-bottom plates (Sarstedt, Numbrecht, Germany) in a volume of 200 μL/well. The culture medium was RPMI 1640 (Sigma-Aldrich) supplemented with 10% fetal calf serum and antibiotics (all from Sigma-Aldrich): penicillin (100 units/mL), streptomycin (0.1 mg/mL), and gentamicin 0.08 mg/mL. The cultures were incubated in a humidified cell incubator at 37 °C with 5% CO_2_. PBMC were treated with PHA (10 μg/mL). For the cytotoxicity assay, the cultures were not treated with PHA. The unstimulated cultures were incubated with double-increasing concentrations of PEx, PG, PN, or EA (1–320 μg/mL) and incubated for 24 h, after which the metabolic activity was determined. PHA-stimulated cultures were treated with double-increasing concentrations of PEx, PG, PN, EA or two different combinations of PG/PN/EA. Combination 1 was: PG 10 μg/mL, PN 20 μg/mL, and EA 1 μg/mL, whereas combination 2 was: PG 20 μg/mL, PN 40 μg/mL, and EA 2.5 μg/mL. Cytokines were analyzed in the supernatants after 3 days, whereas proliferative activity was determined after 4 days. Fresh samples of all compounds were prepared from the original PEx batch or individual ellagitannin components, which were initially dissolved in dimethyl sulfoxide (DMSO). Therefore, PBMC cultures with DMSO alone were used as an additional control. All results of cultures with highest maximal concentrations of DMSO (0.1%) did not significantly differ from the negative control cultures and were not presented (data not shown). The level of endotoxin in PEx at the concentration of 800 μg/mL in the complete culture medium was within the accepted limits (0.92 ng/mL) determined by the Limulus amebocyte lysate (LAL) assay.

### 4.5. MTT Assay

PBMC were cultivated in 96-well plates (triplicates), as described in either fresh complete RPMI medium or the medium with different dilutions of PEx, PG, PN, or EA for 24 h. After that, the plates were centrifuged and the medium was carefully removed. The solution of 3-[4,5-dimethyl-2-thiazolyl]-2,5-diphenyl tetrazolium bromide (MTT) (Sigma-Aldrich) (100 μL/well, final concentration 100 μg/mL) was added to the wells. Wells with different concentrations of the compounds but without cells were used to test the interaction of MTT-developed color with the extract. Wells with an MTT solution without cells served as blank controls. After an 3 h incubation of the plates at 37 °C, formazan crystals were dissolved with 0.1 N HCl/10% sodium dodecyl sulfate (SDS) (100 μL/well) overnight. The optical density (OD) of the developed color was read at 570/650 nm (ELISA reader, Behring II, Marburg, Germany). The results were expressed as the relative metabolic activity compared to the metabolic activity of control cultures. The relative metabolic activity of experimental cultures in relation to control cultures (OD used as 100%) was calculated as follows: metabolic activity (%) = (OD of cultures with tested compounds − OD with the compounds without cells/OD of control cultures − OD of medium without cells) × 100.

### 4.6. Proliferation Assay

PBMCs were labeled with Cell Trace Far Red dye (Invitrogen, Waltham, MA, USA) following the manufacturer’s protocol, then stimulated with PHA with or without different concentrations of PEx, PG, PN, or EA for the next 4 days. After the incubation period, the cells were harvested and stained with Propidium Iodide (PI) (50 μg/mL, Sigma-Aldrich). Cell Trace Far Red dye dilution was analyzed after the exclusion of doublets and PI+ cells by a BD LSRII flow cytometer. The proliferative index (PI), division index (DI) and % of division were determined. The proliferation activity of PBMC in experimental cultures was expressed as relative proliferation to control (mean proliferation in PHA-stimulated cultures without the tested compounds) used as 100%.

### 4.7. Cytokine Measurement

PBMC were incubated for 4 days, as already described. The concentrations of TNF-α, IFN-γ, IL-5, IL-6, IL-8, IL-10, and IL-17A in culture supernatants were detected by specific sandwich ELISA (R&D Systems GmbH, Wiesbaden, Germany) for human cytokines following the instructions of the manufacturer. The concentration of a particular cytokine was determined using a standard curve, based on the known concentrations of cytokines in the ELISA kits. All measurements of cytokines were performed in duplicates, then, the mean values were calculated and expressed as pg/mL. Synergistic effect was recognized if there were statistically significant differences in cytokine concentrations in a culture treated with a combination of PG/PN/EA compared with all three corresponding individual ellagitannin components. If such a difference was observed in relation to two of three components, the effect was considered as an additive.

### 4.8. Statistical Analysis

The Friedman test (paired one-way ANOVA) with Dunn’s multiple comparison post-test was used for comparison of multiple samples. Statistically significant values were *p* < 0.05 and lesser. All analyses were performed with GraphPad Prism 8 (GraphPad Software, La Jolla, CA, USA).

## 5. Conclusions

This study showed that non-cytotoxic concentrations of PG, PN, and EA, the main ellagitannins from PEx, showed a similar dose-dependent effect on T-cell proliferation in human PBMC cultures stimulated with PHA, a T-cell mitogen, in a pattern similar to that observed with PEx. When the effect was compared in relation to the applied concentrations, EA exerted the strongest and PN showed the lowest inhibitory activity. All ellagitannin components exhibited a very strong anti-inflammatory activity, as judged by a dose-dependent reduction of TNF-α, IL-6, and IL-8 levels in culture supernatants. Different combinations of PG, PN, and EA acted synergistically in down-regulating the production of these cytokines. PEx decreased the production of Th1 and Th17 cytokines both at lower and higher concentrations, whereas the effect on Th2 and Treg cytokines was biphasic: stimulatory at lower concentrations and inhibitory at higher concentrations. The effect of all ellagitannins on Th1 and Th17 responses (all concentrations) and Th2 and Treg responses (higher concentrations) was inhibitory, but the synergism between them was not seen. Only lower concentrations of EA augmented Treg response, however, at lower concentrations, neither of the ellagitannin components increased Th2 response. These results imply that most probably other polyphenolic or non-polyphenolic components from PEx are responsible for stimulation of Th2 cells and suggest the complexity of the immunomodulatory activity of PEx, which is more complex than the activity of individual ellagitannin components. The findings could be relevant when using pomegranate or their biologically active compounds for the treatment of autoimmune diseases, allergies, or cancers.

## Figures and Tables

**Figure 1 molecules-27-07871-f001:**
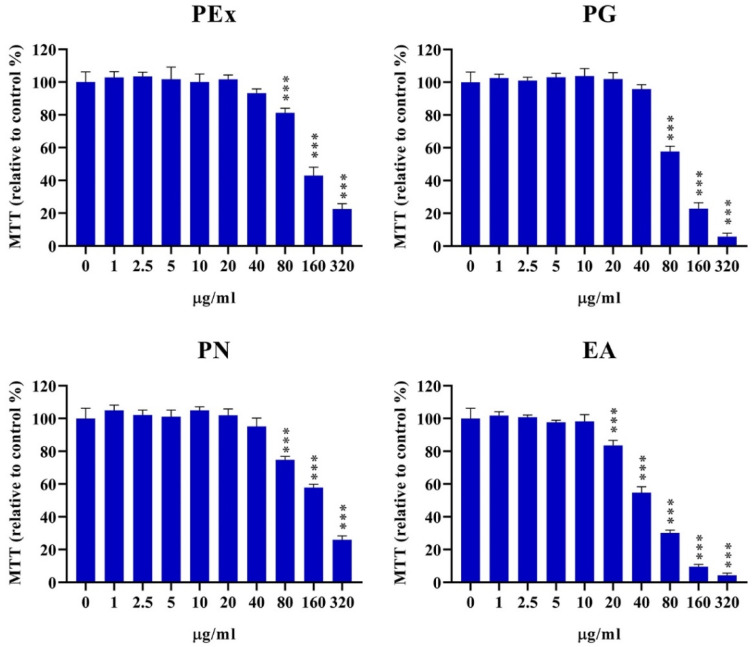
Cytotoxicity of pomegranate peel extract (PEx), punicalagin (PG), punicalin (PN) and ellagic acid (EA) in culture with PBMC. Cytotoxicity of PBMC was determined after 24 using the MTT test as described in Materials and methods. Values are shown as mean ± SD of five donors. *** *p* < 0.005, compared to control non-treated PBMC.

**Figure 2 molecules-27-07871-f002:**
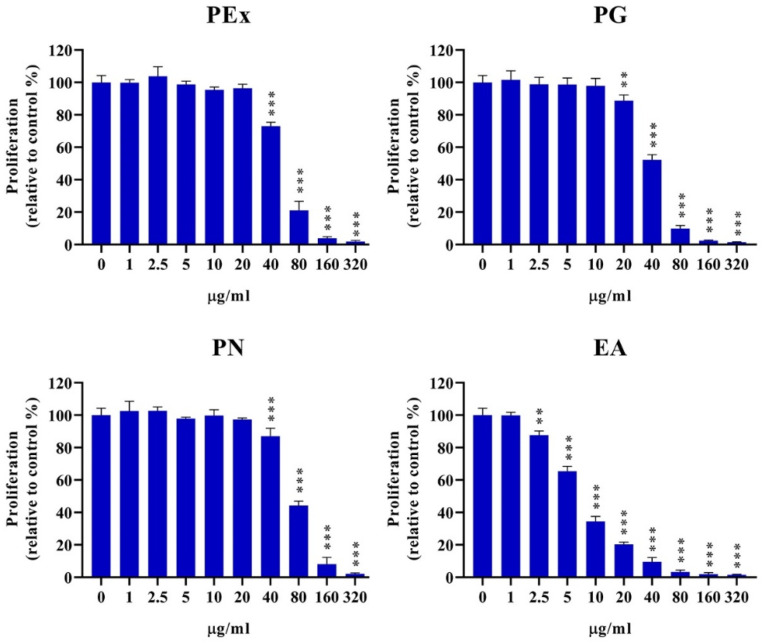
The effects of pomegranate peel extract (PEx), punicalagin (PG), punicalin (PN) and ellagic acid (EA) on PHA-stimulated proliferation of PBMC. PBMC pre-labeled with Cell Trace Far-Red were cultured with different concentrations of the compounds (1–320 μg/mL) or without them in the presence of PHA (10 μg/mL) for 4 days, followed by the analysis of Far-Red dilution by flow cytometry. Values are given as % proliferation relative to control used as 100%. The summarized data are shown as mean ± SD from 5 donors ** *p* < 0.01, *** *p* < 0.005, compared to control non-treated PHA-PBMC.

**Figure 3 molecules-27-07871-f003:**
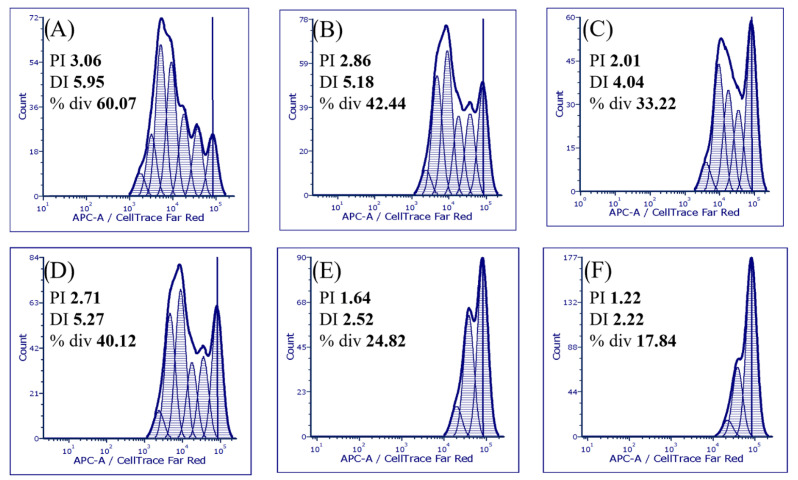
Representative histograms of PBMC proliferation. (**A**) control PHA-stimulated PBMC. (**B**) PEx = 40 μg/mL (**C**) PG = 40 μg/mL (**D**) PN = 40 μg/mL (**E**) EA = 5 μg/mL (**F**) EA = 10 μg/mL. (PI-proliferation index, DI-division index, div-percentage of division).

**Figure 4 molecules-27-07871-f004:**
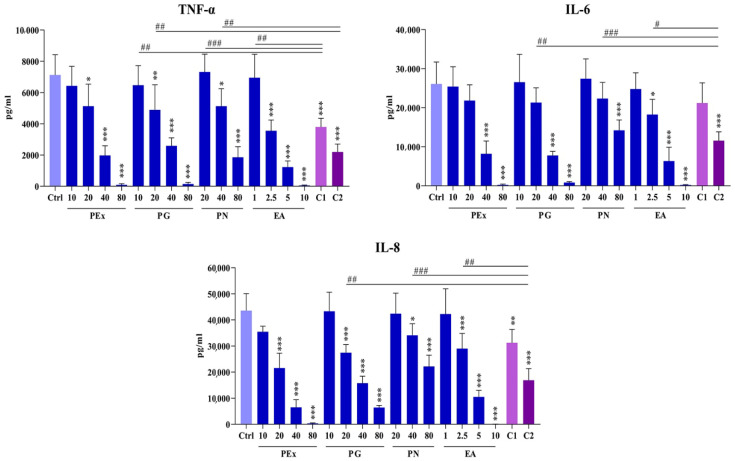
The effects of pomegranate peel extract (PEx), punicalagin (PG), punicalin (PN), ellagic acid (EA), and combinations of these ellagitannin components (C1 and C2) on pro-inflammatory cytokine production in PHA-stimulated PBMC cultures. PBMC cultures were treated with different concentrations of the compounds as indicated or with two different combinations of PG, PN and EA as indicated in Materials and methods. * *p* < 0.05, ** *p* < 0.01, *** *p* < 0.005, compared to control non-treated PBMC. # *p* < 0.05, ## *p* < 0.01, ### *p* < 0.005, compared to indicated individual concentrations of ellagitannins.

**Figure 5 molecules-27-07871-f005:**
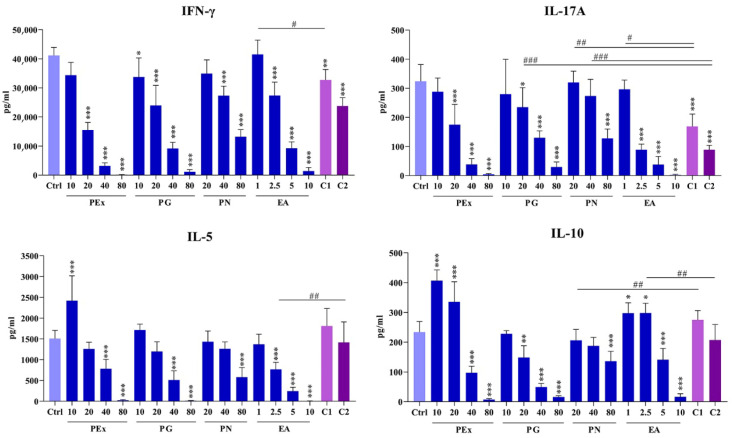
The effects of pomegranate peel extract (PEx), punicalagin (PG), punicalin (PN), ellagic acid (EA) and combinations of these ellagitannin components (C1 and C2) on T-cell cytokine production in PHA-stimulated PBMC cultures. PBMC cultures were treated with different concentrations of the compounds or with two different combinations of PG, PN and EA as indicated in Materials and methods. * *p* < 0.05, ** *p* < 0.01, *** *p* < 0.005, compared to control non-treated PBMC. # *p* < 0.05, ## *p* < 0.01, ### *p* < 0.005, compared to indicated concentrations of individual ellagitannins.

## Data Availability

All data are included in this article.

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
