# Peer review of "Immunomodulatory Activity of Punicalagin, Punicalin, and Ellagic Acid Differs from the Effect of Pomegranate Peel Extract"

_molecules, 2022, doi:10.3390/molecules27227871_

Round 1
Reviewer 1 Report
This referee can’t understand the reason the activities of the different compounds were evaluated by mg/mL. The mmol/L should be employed. The meaning of mg/mL should be explained.
Author Response
This referee can’t understand the reason the activities of the different compounds were evaluated by mg/mL. The mmol/L should be employed. The meaning of mg/mL should be explained.
Response: Since this is a comparative study between the immunomodulatory effect of punicalagin, punicalin, and ellagic acid, the main ellagitannins from the pomegranate, and pomegranate peel extract, it was not possible to express the concentrations in mmol/L because the molecular mass of the extract is not known. In such a case, µg/mL is the only option.
Reviewer 2 Report
The research is interesting, but there are some problems and typos. The manuscript said that the punicalagin, punicalin and ellagic acid were from pomegrantate peel but there is no informations on how the compounds were isolated from the peel. The components of the extract had better analyzed by HPLC-MS. There are too many "-" before the numbers which may resulte in misunderstanding. The temperature "600" (line 397) must be a typo. Line 65 "least" should be "the least". Some references should be better added after Lines 55-56.
Author Response
The research is interesting, but there are some problems and typos. The manuscript said that the punicalagin, punicalin and ellagic acid were from pomegranate peel but there is no informations on how the compounds were isolated from the peel. The components of the extract had better analyzed by HPLC-MS.
Response: We obtained pure components (punicalagin, punicalin, and ellagic acid) from a commercial source as has already been written (pages 407-408). The misunderstanding can be originated from the title of the paper. To avoid this, we changed the title to exclude “from”. The suggested title now is Immunomodulatory Activity of Punicalagin, Punicalin, and Ellagic Acid Differs from the Effect of Pomegranate Peel Extract. We think that HPLC is enough (already published in reference 24, and supplementary Fig.1), because the components were not isolated from the extract.
There are too many "-" before the numbers which may resulte in misunderstanding.
Corrected (marked in red) (lines 103-104, 119, 132)
The temperature "600" (line 397) must be a typo.
Corrected (marked in red)
Line 65 "least" should be "the least".
Corrected (marked in red)
Some references should be better added after Lines 55-56.
Corrected (marked in red)
Round 2
Reviewer 1 Report
This referee can’t evaluate the value this article. Because the assay was shown by mg. The mol/L should be employed.